# Altered Development of Prefrontal GABAergic Functions and Anxiety-like Behavior in Adolescent Offspring Induced by Prenatal Stress

**DOI:** 10.3390/brainsci12081015

**Published:** 2022-07-31

**Authors:** Arbthip Suwaluk, Nuanchan Chutabhakdikul

**Affiliations:** Research Center for Neuroscience, Institute of Molecular Biosciences, Mahidol University, Nakhon Pathom 73130, Thailand; p.arbthip@gmail.com

**Keywords:** prenatal stress, prefrontal cortex, adolescence, GABAergic functions, anxiety

## Abstract

Maternal stress can afflict fetal brain development, putting the offspring at risk of cognitive deficits, including anxiety. The prefrontal cortex (PFC), a protracted maturing region, is notably affected by prenatal stress (PS). However, it remains unclear how PS interferes with the maturation of the GABAergic system, considering its functional adjustment in the PFC during adolescence. The present study thus investigated the long-lasting consequences of PS on the prefrontal GABAergic functions of adolescent offspring. Pregnant Sprague–Dawley rats were divided into controls and the PS group, which underwent restraint stress during the last week of gestation. Male pups from postnatal days (PND) 40–42 were submitted to the elevated plus maze (EPM) test. Proteins essentially involved in GABAergic signaling were then examined in PFC tissues, including the K^+^-Cl^−^ cotransporter (KCC2), Na^+^-K^+^-Cl^−^ cotransporter (NKCC1), α1 and α5 subunits of GABA type A receptors (GABA_A_ receptors), and parvalbumin (PV), along with cAMP response element-binding protein phosphorylation (pCREB), which reacts in the plasticity regulation of PV-positive interneurons. The results revealed that the higher anxiety-like behavior of PS adolescent rats concurred with the significant decreases of the KCC2 and α1 subunits, with PV- and pCREB-lowered levels. The findings suggested that PS disrupts the continuance of PFC maturity by reducing the essential elements of GABAergic functions. These changes likely underlie the anxiety emerging in adolescence, possibly progressing to mental disorders.

## 1. Introduction

Maternal stress hormones negatively impact fetal growth and cause long-term consequences on the brain and behavioral development of the offspring [1,2]. Considering the brain areas crucial for cognitive and emotional control, the prefrontal cortex (PFC) is one of those markedly affected by prenatal stress (PS), especially at the neuronal and synaptic levels [3,4]. In addition, the PFC is an extended developing region and highly plastic during adolescence, which is a period for the emergence of neuropsychiatric diseases [5,6], including anxiety disorders [7,8]. The vulnerability to such diseases notably connects with the prefrontal GABAergic system disruptions [9,10].

The essential elements of the neurotransmitter GABA (γ-aminobutyric acid) signaling include cation-chloride cotransporters (CCCs), which actively regulate the intracellular chloride (Cl^−^) concentration [11]. The developmental switching of GABAergic functions from excitation to inhibition depends on the upregulation of the Cl^−^ exporter, K^+^-Cl^−^ cotransporter (KCC2) rather than the Cl^−^ importer, Na^+^-K^+^-Cl^−^ cotransporter (NKCC1), which occurs during the early postnatal period in rodents, concurrent with the regulation of the intraneuronal Cl^−^ level [12,13,14]. In addition, anion-permeable channels, GABA type A receptors (GABA_A_ receptors), are another vital component in GABAergic functions; GABA_A_ receptors establish and maintain the Cl^−^ gradient in coordination with KCC2 and NKCC1 [11,15]. In the PFC, a developmental rise in GABA_A_ receptor α1 subunits corresponds to a reduction in α5 subunits expression with increasing age until maturity [16,17]. Accordingly, α1 subunits incorporate short-lasting responses, conducting fast phasic inhibition [18,19], while α5 subunits involve slow tonic inhibitory conductance [20,21].

Previous studies suggested that PS interferes with CCCs and GABA_A_ receptor subunits, then interrupts the maturity of GABAergic functions, leading to aberrant behaviors. For instance, in the hippocampus (HPC) of rat pups at puberty, PS delayed the GABAergic system maturation by changing the expression of NKCC1 and KCC2, along with GABA_A_ receptor α1 and α5 subunits [22]. Additionally, by measuring the gene expression in adolescent rats, they found that PS primarily increased the NKCC1/KCC2 ratio in the HPC associated with reduced social behaviors [23], as in the amygdala (AMG), PS decreased the expression of KCC2 and increased NKCC1, enhancing the anxiety-like behavior of females [24]. Despite this previous evidence, how PS alters the development of the prefrontal GABAergic system, specifically in adolescence, remains unclear. However, it is somehow related to a significant adjustment of the local GABAergic functions in the PFC during adolescence to drive the maturation of the excitatory/inhibitory (E/I) balance in the prefrontal network [9]. Furthermore, PS has developmental consequences on the substantial group of GABAergic interneurons that express parvalbumin (PV), one of the calcium (Ca^2+^)-binding proteins markers [25]. PS affected PV-positive interneuron proportion leading to delayed maturation in the HPC and medial frontal cortex (mFC) from childhood to adulthood, correlated with anxiety-like behavior in adult mice [26]. Apart from changes at the cellular level, it should be in question whether PS influences the proteins associated with the molecular mechanism of interneurons. Interestingly, the meticulous study of PV-positive cortical interneurons demonstrated a novel Ca^2+^/calmodulin-dependent kinase (CaMK) pathway, triggering cAMP response element-binding protein phosphorylation (pCREB) in activity-dependent gene regulation [27]. Thus, a better understanding of PS effects related to long-term plasticity would provide by investigating PV coupled with pCREB.

This present study aims to examine the effects of PS on the developmental trajectories of the GABAergic system, i.e., the protein levels of KCC2, NKCC1, GABA_A_ receptor α1/α5 subunits, PV, and pCREB, during adolescence in the PFC of male rat pups because of a greater tendency of anxiety-like behavior in males evoked by prenatal psychological stressors such as maternal restraint stress [28]. The information contributed by this study can expand the evidence of PS-induced cellular and molecular alterations in the prefrontal GABAergic functions, which may underlie the anxiety-like behavior in adolescent offspring.

## 2. Materials and Methods

This study was conducted according to the Guidelines for Care and Use of the Laboratory Animals and approved by the Institute ACUC Committee (COA.NO. MB-ACUC 2015/003).

### 2.1. Animals

Pregnant Sprague–Dawley rats were transported on the expected gestational day (GD) from the National Experimental Animal Center of Mahidol University. Each was housed individually in a cage with 12 h of light/dark cycle, temperature and humidity control, and free access to food and water. Rats were weighed daily before any manipulation. Then, each pregnant female received nesting materials on the designated delivery day, and the cages were checked twice daily. The appearance of a litter marked postnatal day (PND) 0; pups were weaned on PND 21.

### 2.2. Maternal Restraint Stress

After random division into the control and PS groups (*n* = 4), pregnant rats in the PS group underwent immobilization with bright light in the well-ventilated plexiglass cylindrical cage with adjustable lengths. Restraint stress was carried out for four hours daily from GD 14 to GD 20 during the light phase of the cycle [22,29]; the observation of restlessness or suffering signs happened every half an hour. For the control group, pregnant rats were left undisturbed until parturition. The operation of maternal restraint stress was in the last week of gestation, because this period presents the most sensitivity to the behavioral teratogenic effect of PS [30]. Moreover, epigenetic modifications grow more complicated in the late gestational stage, and PS can disturb synaptic remodeling in the brain during this time [3].

### 2.3. Elevated plus Maze

The elevated plus maze (EPM) test occurred in the dark phase (rats’ active period), and rat behaviors were recorded and analyzed by the SMART video tracking system version 3 (Panlab, Harvard, MA, USA). The apparatus consisted of a plus-shaped maze elevated 50 cm above the floor with two opposite closed arms and open arms (50 × 10 cm) and a center platform (10 × 10 cm). We performed the test in a dark room under dim light to prompt open arms entry of the controls, as suggested by Weinstock [2]. On PND 40–42, male pups from both groups (*n* = 5) were taken separately from their home cages and rested briefly in transport cages near the testing area. When the record started, each rat stayed on the central platform facing an open arm; the record ran for five minutes, while the observer monitored outside the testing room to avoid any disturbances. Then, the rat was removed from EPM and returned to its home cage before bringing the next one into the room. We thoroughly cleaned the maze floor with diluted citric acid and 70% alcohol between each test. The amount of time spent in open and closed arms, the number of entries into each arm, and the distance traveled in both arms were calculated for anxiety-like behavior assessment [31,32].

### 2.4. Tissue Preparation

On PND 43–45, rat pups from both groups were decapitated, and fresh PFC tissues were dissected and immediately stored at −80 °C for further use. For fixation, intact pieces of the PFC were immersed in 4% paraformaldehyde in 0.1 M PBS, pH 7.4. The incubation was two nights at 4 °C, then cryoprotection with 30% sucrose in 0.1 M PBS, pH 7.4. Cryoprotected tissues were embedded in OCT media to set the cutting block and coronally cut for 30 µm-thick in the cryostat machine, hence ordered as ten serial coronal sections. Brain slices were mounted on the gelatin-coated glass slides and kept in the container under −20 °C for further use.

### 2.5. Reagents

The antibodies were purchased from Santa Cruz Biotechnology (Dallas, TX, USA) as follows; polyclonal goat anti-KCC2 (SC-19420), polyclonal goat anti-NKCC1 (SC-21545), polyclonal goat anti-PV (SC-7448), polyclonal goat anti-GABA_A_ α1 (SC-7348), polyclonal goat anti-GABA_A_ α5 (SC-7357), and mouse anti-β-actin (SC-69879), including the HRP-conjugated secondary antibodies. Additionally, the monoclonal rabbit anti-pCREB (87G3), polyclonal rabbit anti-tubulin β (SAB4500088), and polyclonal rabbit anti-KCC2 (07-432), were obtained from Cell Signaling Technology (Danvers, MA, USA), Sigma-Aldrich (St. Louis, MO, USA), and MilliporeSigma (Burlington, MA, USA), respectively. Donkey anti-rabbit IgG–secondary antibody (Alexa Fluor 594) and the nuclear dye, DAPI, were purchased from Invitrogen, Thermo Fisher Scientific (Waltham, MA, USA). Lastly, the enhanced chemiluminescence (ECL) reagents (Amersham ECL Prime) were obtained from GE Healthcare (Chicago, IL, USA) and the anti-fade mounting solution (VECTASHIELD) was purchased from Vector Laboratories (Newark, CA, USA).

### 2.6. Western Blot Analysis

The supernatants from PFC tissue lysate were processed through concentration measurement using the Bradford Assay. After concentration adjustment and protein denaturation, each sample with equal volume was resolved in 10–12% SDS-PAGE for 120 minutes before transferring to the PVDF membrane at 100 V for 120 minutes. The membranes were blocking-incubated in 3% bovine serum albumin (BSA) in Tris-buffered saline containing 0.1% Tween-20 (TBST) for the detections of KCC2, NKCC1, PV, pCREB, GABA_A_ receptor α1, and α5 subunits, and in 5% skimmed milk buffer for β-Actin and Tubulin β. Next, the membrane incubation of primary antibody diluted in TBST was applied overnight at 4 °C with antibodies from the commercial sources as follows: polyclonal goat anti-KCC2 (1:1000), polyclonal goat anti-NKCC1 (1:1000), polyclonal goat anti-PV (1:1000), monoclonal rabbit anti-pCREB (1:1000), polyclonal goat anti-GABA_A_ receptor α1 subunit (1:500), polyclonal goat anti-GABA_A_ receptor α5 subunit (1:1000), mouse anti-β-actin (1:10,000), and polyclonal rabbit anti-tubulin β (1:10,000). After that, the membranes were washed for five minutes each, three times in TBST, and incubated at room temperature with compatible HRP-conjugated secondary antibodies diluted in TBST. The membranes were then incubated with the ECL reagents for protein band signaling and detected by film capture or the imaging system machine (Azure Biosystems, Dublin, CA, USA). Band densities were quantified with the ImageJ program developed by the National Institutes of Health, USA.

### 2.7. Immunohistochemistry

Brain sections were blocking-incubated at room temperature with buffer containing 10% donkey serum plus 0.2% Triton X-100 in 0.1 M PBS, pH 7.4. Next, the sections were incubated with the diluted primary antibody, i.e., polyclonal rabbit anti-KCC2 (1:500) (MilliporeSigma, Burlington, MA, USA) in 2% donkey serum plus 0.2% Triton X-100 in 0.1 M PBS, pH 7.4 buffer, overnight at 4 °C. After that, the sections were washed in 0.1 M PBS, pH 7.4, for five minutes each, three times. The sections were then incubated with donkey anti-rabbit IgG–secondary antibody (Alexa Fluor 594; 1:500) (Thermo Fisher Scientific, Waltham, MA, USA) and the nuclear dye (DAPI; 1:1,000), diluted in the buffer same as the primary antibody incubation for 1 to 2 hours in a dark-humid chamber at room temperature. After thorough washing, stained brain sections were maintained by the anti-fade mounting solution.

### 2.8. Image Analysis

The fluorescent-stained brain sections were imaged under a ZEISS confocal microscope (Carl Zeiss Microscopy, Jena, Germany) to scan through the z-plane at 40× magnification. The same parameters were applied to all imaging of each section from both groups to control the signal intensity. Two-dimensional images were adjusted with the maximum projection by ZEN microscopy software (ZEN 2.1, Carl Zeiss Microscopy, Jena, Germany)

### 2.9. Statistical Analysis

The data were presented as the mean ± SEM and statistically analyzed using GraphPad Prism version 5 software (GraphPad Software Inc., San Diego, CA, USA). Accordingly, protein quantification data, the percentage of time spent, the number of entries, and the percentage of distance in the EPM test were compared between the PS and control groups using a Student’s *t*-test (unpaired two-sample *t*-test). A *p*-value ≤ 0.05 was considered significant between the two groups.

## 3. Results

### 3.1. Anxiety-like Behavior in Adolescent Rats

PS increases the risk for mental illness in the offspring, such as anxiety [4]. We then assessed anxiety-like behavior in rats based on behaviors in response to a novel set in the EPM, reflecting a conflict between their innate need to explore new surroundings and liking for a dark-enclosed space, which can provoke an avoidance of a high-open area [31,32]. Decreased time spent in the open arms and/or reduced number of entries into the open arms indicate anxiety, while the number of closed arms entries indicates spontaneous motor activity [31], as well as the distance traveled in each area was also measured [32]. As a result, the percent of time spent in the open arms (Figure 1A) was significantly lower in PS adolescent pups than controls, consistent with more time spent in the closed arms (Figure 1B), suggesting higher anxiety in PS male rats. However, the number of entries into the open arms (Figure 1C) and closed arms (Figure 1D), also, the distance traveled in the open arms (Figure 1E) and closed arms (Figure 1F) showed non-significance, implying no difference in the locomotor activity between groups.

### 3.2. Developmental Shift of Cation-Chloride Cotransporters

KCC2 upregulation is essential for maintaining neuronal Cl^−^ homeostasis in GABA inhibitory actions [14]. To investigate this component as a part of GABAergic maturation, Cl^−^ exporter, KCC2, and Cl^−^ importer, NKCC1, were measured at the protein levels. As a result, PS significantly decreased KCC2 (Figure 2B), and the deficiency of KCC2 expression also showed in the immunohistochemical staining of the medial PFC (Figure 2A). However, there was no significant difference in the NKCC1 level compared to the controls (Figure 2C). These results indicate altered KCC2 switching in the PFC of PS adolescents.

### 3.3. GABA Type A Receptors Functional Development

Fast phasic inhibition in GABAergic signaling intrinsically contains GABA_A_ receptor α1 subunits; with increasing age in the PFC, an upregulation in α1 subunits aligns with α5 subunit reduction [16,17,18,19]. The protein levels of the α1 and α5 subunits were then measured to investigate the maturing expression of these subunits regarding the functionality of GABAergic inhibition. The results demonstrated PS induced a significant decrease in α1 subunits (Figure 3A) but not α5 subunits (Figure 3B), indicating impaired α1 subunits shifting in the PFC of PS adolescent rats.

### 3.4. The Plasticity Regulation of GABAergic Interneurons

PV-positive/fast-spiking cells, the substantial group of cortical interneurons, retain prolonged development through the young periods until adolescence [25]. PV is a Ca^2+^-binding protein, and Ca^2+^ dynamics in PV-positive cells involve the Ca^2+^-dependent signaling pathway that regulates the gene expression necessary for long-term plasticity by activating pCREB [27]. The protein levels of PV and pCREB were thus measured to inspect the development of interneurons and the plasticity regulation that can affect their functions. As a result, PS significantly decreased both PV (Figure 4A) and pCREB (Figure 4B), suggesting altered PV and pCREB-dependent activities in the PFC of PS adolescents.

## 4. Discussion

Maternal stress during pregnancy has been extensively studied for long-lasting effects on the offspring, predisposing cognitive dysfunctions later in life [2,4]. It is still unclear how PS alters the development of prefrontal GABAergic functions in adolescence, inducing impairments such as anxiety. Thus, the present study investigated PS-induced alterations in the elements of the GABAergic system in the PFC of adolescence regarding anxiety in the offspring. The results showed a decrease in KCC2 protein expression in PS rats, while there was no significant difference in the level of NKCC1, indicating an altered developmental shift of KCC2. Moreover, a reduction of GABA_A_ receptor α1 subunits suggests changes in GABA_A_ receptors functionality, although the level of α5 subunits showed no difference. PS also induced alterations of PV and pCREB-dependent activities in the PFC by decreasing both the PV and pCREB protein levels. Therefore, these consequences disturb the prefrontal GABAergic functions, which likely underlies anxiety-like behavior in prenatally stressed adolescent males. However, there was a limitation on the age of adolescence in our study that only tested rats’ behavior on PND 40–42 and collected their brain tissues on PND 43–45. Accordingly, this selected age range corresponds to middle adolescence (PND 34–46), which rodents show increased impulsivity and risk-taking, as reviewed by Holliday and Gould [33]. The range between PND 30 and PND 60 in rats and mice is comparable to adolescence in other species related to behavioral and neurobiological changes, despite no clear-cut point at what age span covers the adolescent period [34]. In addition, rats/mice at around PND 60 are conventionally assessed as adults when reaching physical and sexual maturity [35].

Recently, Milani et al. [28] have proposed that PS-precipitated anxiety-like behaviors result from an imbalance in GABAergic and glutamatergic circuits directly or indirectly induced by neuroinflammation. Both GABAergic and glutamatergic transmission in the PFC undergo significant arrangement during adolescence, particularly GABAergic signaling, which plays a crucial role in refining glutamatergic activity and cortical maturation [6,9,36]. Prefrontal adjustment in adolescence points out the high plasticity of the PFC, and disruptions during this susceptible period led to impaired maturation of the prefrontal GABAergic circuits in adulthood [37,38]. According to anxiety, the modulatory model has demonstrated an imbalance between overactive bottom-up activity from the AMG and dysfunctional top-down control emerging in the PFC [39]. In addition, collecting data in neuroimaging studies has suggested the importance of cortical–subcortical regulatory mechanisms, especially the medial PFC hypoactivity in anxiety disorders [40]. Thus, the integrity of the PFC is necessary for emotional control, and the GABAergic system development in this region is worth understanding.

Focusing on the key elements of GABA transmission, our study examined how PS disturbs the prefrontal development of GABA_A_ receptors by investigating α1 and α5 subunits, since, with increasing age in the PFC, an upregulation in α1 subunits coincides with α5 subunit reduction until adulthood [16,17]. It is known that GABA_A_ receptors mediate fast phasic and slow tonic inhibition by synaptic and extrasynaptic signaling, respectively [11,21]. The extrasynaptic GABA_A_ receptors integrated α5 subunits that downregulate during development [20], as opposed to α1 subunits that promoted faster decay times or rapid responses [18,19]. A developmental shift from α2- to α1-containing GABA_A_ receptors in the PFC also induced faster inhibitory kinetics during adolescence [41]. In this time window, the increased frequency of inhibitory postsynaptic currents (IPSCs) onto pyramidal neurons and GABA_A_ receptor-mediated responses to afferent drive recorded by local field potential manifested the enhancement in prefrontal GABAergic functions [9,42]. Accordingly, homozygous α1^−/−^ mice lacking the primary adult GABA_A_ α1 subunits continued a juvenile phenotype of IPSCs with longer decay time constant values than the wild-type mice, holding the immature form of postsynaptic responses in the cortex [43]. Hence, PS can profoundly impact the maturation of prefrontal GABAergic signaling by interrupting developmental switching of GABA_A_ receptor α1 subunits, as found in our study.

The adolescent enhancement of GABAergic functions in the PFC also connects with an increase in the total PV levels due to the distribution of PV-positive processes with a slight increase in PV-positive cells [44]. However, the reduced PV protein level in our study probably relates to the delayed migration of cortical GABAergic progenitors caused by PS [45]. The delayed sequence continued after birth, with a decreased population of PV-positive cells in the HPC and mFC in adolescence; despite no differences in total GABAergic cells in adulthood, an altered proportion of PV-positive interneurons underlie anxiety-like behavior [26]. Moreover, prenatal exposure to a proinflammatory cytokine, interleukin-6, could repeat the effect of PS on interneuron progenitor migration [46]; additionally, in the maternal immune activation model, disruptions of PV-positive interneurons in the PFC correlated with anxiety-like behavior in the adult offspring [47]. Therefore, either alone or combined with immune activation, PS can affect GABAergic interneuron development, leading to an imbalance in GABAergic and glutamatergic circuits that precipitate anxiety [28], as mentioned earlier.

We also showed that PS reduced pCREB, an activated transcription factor in a novel CaMK-dependent pathway that controls the plasticity of PV-positive interneurons [27]. As one of the neuronal Ca^2+^-binding proteins, the prevalence of PV suggests that Ca^2+^ dynamics regulation is vital for the activity of these cells [48]. Consistently, PV-positive interneurons are fast-spiking neurons and potent regulators for the E/I balance in the PFC, mediating gamma oscillations [49]. In schizophrenia, MRI-based investigations measuring cortical GABA offered evidence of decreased GABA transmission linked with abnormal gamma-band activity [50]. Additionally, the anxiety disorders rates were significantly higher in schizophrenia patients than in the general population [51], and PS is well-known to be a risk factor for psychiatric disorders, including schizophrenia and anxiety involved with inhibitory neuron pathology [52]. Thus, decreased protein levels of PV and pCREB, possibly leading to altered PV-dependent activity, play a part in PS-induced deficits of inhibitory functions. However, there was another limitation in this study, because we did not directly investigate pCREB in PV-positive interneurons. Therefore, altered plasticity regulations in this cell type need more specific inspections in future studies.

Furthermore, we demonstrated that PS causes long-term consequences by interfering with the developmental shift of KCC2 in the PFC. In the mature brain, the intracellular Cl^−^ concentration is typically low to support hyperpolarization mediated by Cl^−^ influx through GABA_A_ receptors [14,53]. The gene-targeted KCC2^−/−^ mice model highlighted the significance of KCC2 for neuronal Cl^−^ homeostasis and cortical GABA transmission, in which cortical neurons lacking KCC2 failed in a developmental decrease of intracellular Cl^−^ and Cl^−^ loading regulations [54]. In addition, hypomorphic KCC2-deficient mice also showed various deficits, including increased anxiety-like behavior [55]. Additionally, the gene expression study reported reduced social behaviors in prenatally stressed adolescent rats with decreased brain-derived neurotrophic factor (Bdnf) transcripts and an increased NKCC1/KCC2 ratio in the HPC, but in the PFC, there were only gender-dependent changes in Bdnf [23]. Nonetheless, our study showed decreased protein levels of KCC2 in the PFC of rats at the same age period. These inconsistent findings are possibly associated with the different effects of PS on transcriptional and post-translational controls of KCC2 expression [56]. Transforming growth factor β2 (TGF-β2) has been introduced as a post-translational regulator of KCC2 for membrane trafficking and functionality by activating pCREB and Ras-associated binding protein 11b in its pathway [57]. Interestingly, the finding of pCREB in our study corresponds with this cascade; we then suggest a further question of whether the decrease in pCREB induced by PS is related to TGF-β2 signaling that consequently disturbs KCC2 expression.

In conclusion, PS elicits long-term effects by interrupting the maturity process of the GABAergic system, leading to disruptions of the prefrontal E/I balance underlying adolescent anxiety that can further progress to mental illness. Accordingly, the adolescent period is a transition from being children to adults, a manifestation of disturbances in the continuance of PFC maturity, as shown in our study, anticipating the problems of emotional and cognitive control in adulthood. Therefore, more understanding of how PS affects adolescent brains and behaviors is necessary to prevent neuropsychiatric diseases and improve the treatments. In addition, future studies may need to clarify the mechanisms that these studied factors contribute to alterations in the developmental trajectories of the PFC and provide the prospects for intervention targets.

## Figures and Tables

**Figure 1 brainsci-12-01015-f001:**
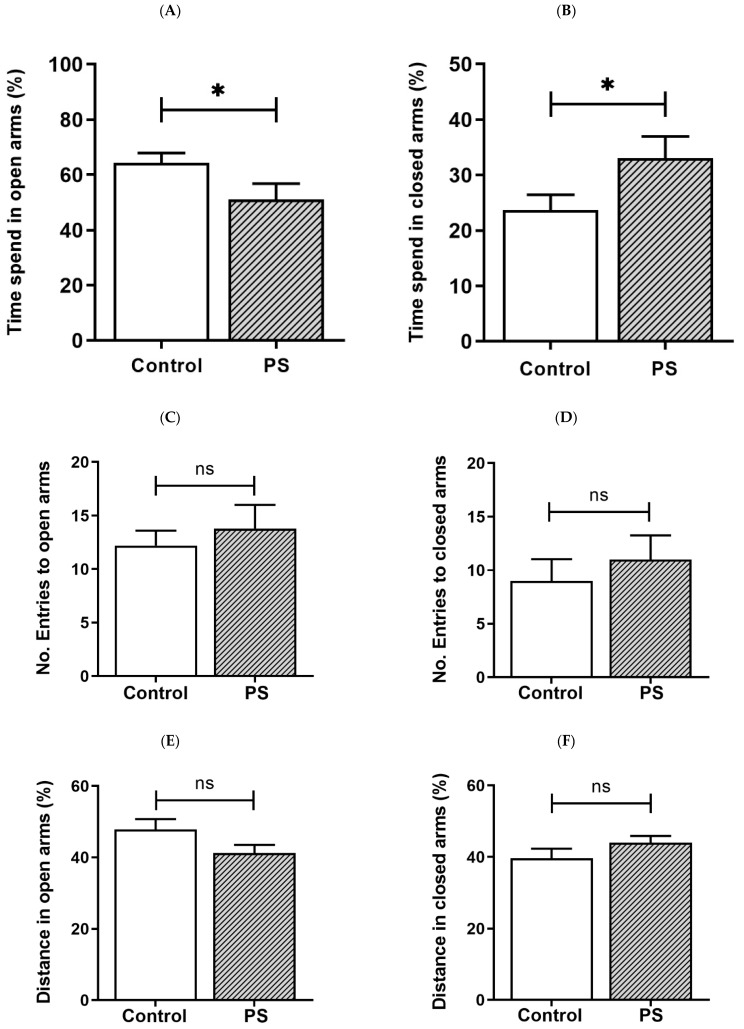
Anxiety-like behavior in adolescent rats. (**A**) Time spent in the open arms. (**B**) Time spent in the closed arms. Bar graphs display the results from the percentage calculation compared between the control and prenatal stress (PS) groups; the significant differences are at * *p* < 0.05. (**C**) Number of entries into the open arms. (**D**) Number of entries into the closed arms. Bar graphs display the results from counted number calculation compared between the control and PS groups; non-significance (ns) was found. (**E**) Distance in the open arms. (**F**) Distance in the closed arms. Bar graphs display the results from the percentage calculation compared between the control and PS groups; non-significance (ns) was found. Data are shown as the mean ± SEM, *n* = 5.

**Figure 2 brainsci-12-01015-f002:**
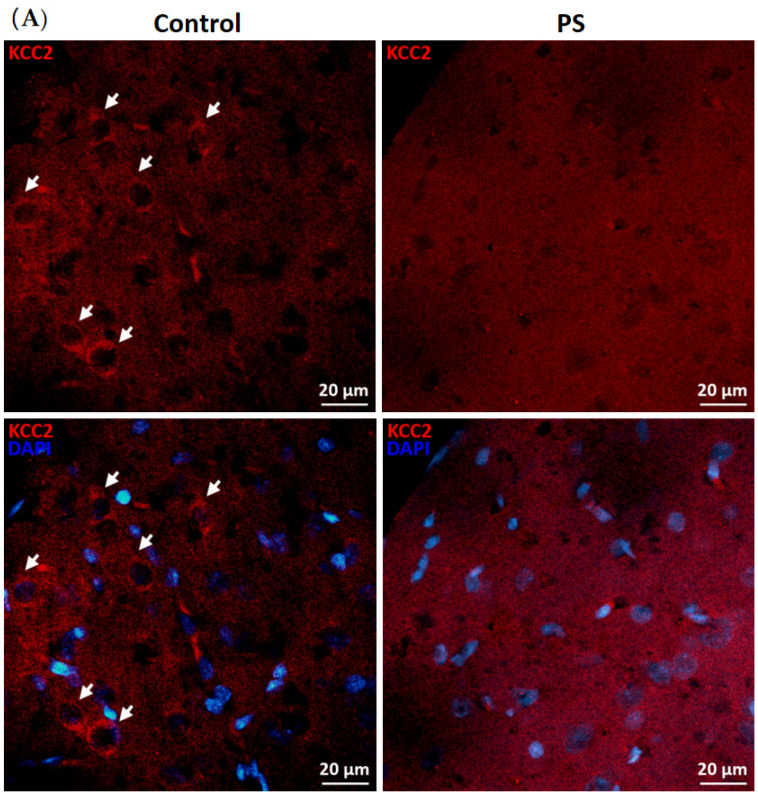
Developmental shift of cation–chloride cotransporters. (**A**) Immunohistochemical staining of KCC2. Expression in the medial prefrontal cortex between the control (left) and prenatal stress (PS) (right) groups; white arrows indicate KCC2-positive cells (upper; red) merged with the nuclear stain DAPI (lower; blue), scale bar = 20 µm. (**B**) Western blotting of KCC2. Upper, band comparing the control (left) and PS (right) groups; lower, bar graph displays the analysis of protein band densities of the KCC2/ß–actin ratio; the significant differences at *** *p* < 0.001. (**C**) Western blotting of NKCC1. Upper, band comparing the control (left) and PS (right) groups; lower, bar graph displays the analysis of protein band densities of the NKCC1/ß–actin ratio; non-significance (ns) was found. Data are shown as the mean ± SEM, *n* = 5.

**Figure 3 brainsci-12-01015-f003:**
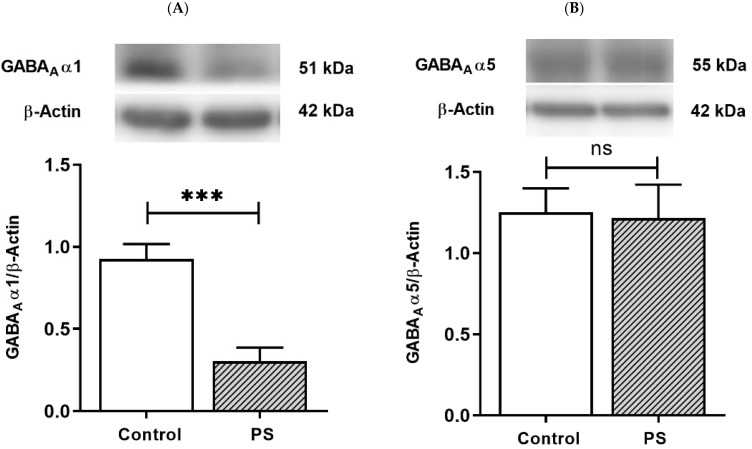
GABA type A receptors functional development. (**A**) Western blotting of GABA_A_ receptor α1 subunits. Upper, band comparing the control (left) and prenatal stress (PS) (right) groups; lower, bar graph displays the analysis of protein band densities of the GABA_A_ receptor α1/ß–actin ratio; the significant differences at *** *p* < 0.001. (**B**) Western blotting of the GABA_A_ receptor α5 subunits. Upper, band comparing the control (left) and PS (right) groups; lower, bar graph displays the analysis of protein band densities of the GABA_A_ receptor α5/ß–actin ratio; non-significance (ns) was found. Data are shown as the mean ± SEM, *n* = 5.

**Figure 4 brainsci-12-01015-f004:**
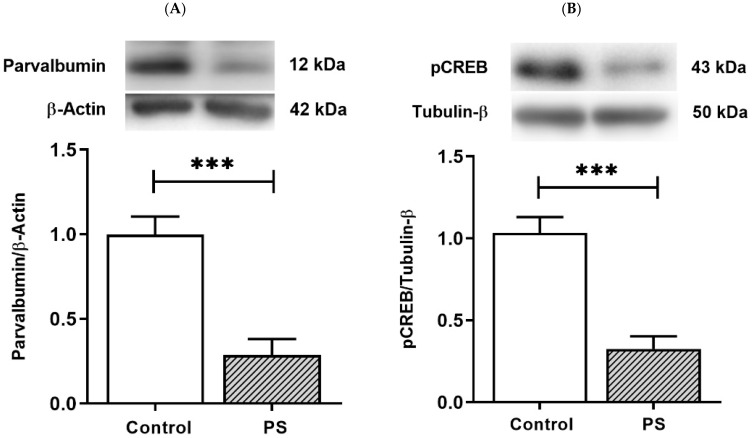
The plasticity regulation of GABAergic interneurons. (**A**) Western blotting of PV. Upper, band comparing the control (left) and prenatal stress (PS) (right) groups; lower, bar graph displays the analysis of protein band densities of the PV/β–actin ratio; the significant differences at *** *p* < 0.001. (**B**) Western blotting of pCREB. Upper, band comparing the control (left) and PS (right) groups; lower, bar graph displays the analysis of protein band densities of the pCREB/Tubulin-β ratio; the significant differences at *** *p* < 0.001. Data are shown as Mean ± SEM, *n* = 5.

## Data Availability

Not applicable.

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
