# Peer review of "Altered Development of Prefrontal GABAergic Functions and Anxiety-like Behavior in Adolescent Offspring Induced by Prenatal Stress"

_brainsci, 2022, doi:10.3390/brainsci12081015_

Round 1
Reviewer 1 Report
This work thoroughly investigates the effect of prenatal stress on prefrontal cortex GABA related molecular elements. The justification for the work is well explained and a comprehensive literature review of previous work in this area is presented in a clear manner.
The results provide convincing evidence of specific changes of proteins related to GABAergic mechanisms in the prefrontal cortex, and thus provide a useful contribution to the current knowledge of the effect of prenatal stress on brain development
One possible limitation of the work is that the authors only look at one time point in the pups development. This should be highlighted in the discussion and the justification for the choice of timing of sacrificing the pups could be further discussed especially as the authors are looking at a period of active brain development.
Overall this is an interesting research that contributes to current knowledge.
Reviewer 2 Report
The authors investigated the role of GABAergic system in the prefrontal cortex in anxiety induced by maternal stress in adolescent Spraque Dawley rats. Maternal stress was developed during the last week of gestation, and male pups were subjected elevated plus maze at the age of 40-42 postnatal days. Furthermore, biochemical and molecular analysis was undertaken. Results revealed higher anxiety-like behavior in stressed adolescent rats and significant decreases of KCC2 and alpha1 subunits with PV and PCREP lowered levels.
The manuscript is interesting and well-written. The methods are well-described and the results correctly shown. However, I have one question concerning the elevated plus maze test. First - whether the experiments were performed in a dark or light room. Moreover, why were the results shown only as time spent in open arms and in closed arms? Usually, as parameters in this test, times actually spent in open arms is counted, but also the number of entries into the open arms. In this test, very important is the locomotor activity of the animals. The authors did not show this parameter - and it is very important. So the authors should introduce such data.
